# Psychometric Qualities Evaluation of the Interdependent Happiness Scale across Malaysia, Philippines, and India

**DOI:** 10.3390/ijerph19010187

**Published:** 2021-12-24

**Authors:** Chee-Seng Tan, Shue-Ling Chong, Argel Bondoc Masanda, Sanju George

**Affiliations:** 1Department of Psychology and Counselling, Faculty of Arts and Social Science, University Tunku Abdul Rahman (UTAR), Kampar 31900, Malaysia; 2Department of Psychology, School of Medicine, International Medical University, Kuala Lumpur 57000, Malaysia; ChongShueLing@imu.edu.my; 3Guidance Services Unit, Office of Student Affairs, Central Luzon State University, Science City of Munoz 3119, Nueva Ecija, Philippines; argelmasanda@clsu.edu.ph; 4Lisie Hospital, Kochi 682025, Kerala, India; sanjugeorge531@gmail.com

**Keywords:** Asia, factor structure, happiness, interdependent happiness

## Abstract

The nine-item Interdependent Happiness Scale (IHS; Hitokoto & Uchida, 2015) is a self-report of interpersonal happiness that focuses on three dimensions: relationship-oriented happiness, quiescent happiness, and ordinary happiness. Few studies have evaluated the psychometric properties of the IHS in diverse cultural backgrounds and the findings are inconsistent. This study investigated whether the IHS has sound psychometric qualities in three Asian countries. University students from Malaysia (*n* = 263), Philippines (*n* = 239), and India (*n* = 310) answered the IHS and self-rated creativity scale. Confirmatory factor analysis on each sample supported the nine-item second-order model with error covariances. The overall IHS score showed good reliability in all samples. The subscales, however, had mixed results except for the Indian sample. Similarly, the convergent validity test showed mixed results while discriminant validity is supported in all samples except for the quiescent happiness subscale in the Indian sample. Concurrent validity was established across three samples by showing a positive relationship with creativity score. The results highlight that the higher-order structure of the IHS is consistently supported in different cultural contexts. However, some of the items are perceived differently and require further improvement in enhancing the cross-cultural usability of the IHS to measure socially-oriented happiness.

## 1. Introduction

Happiness is a desirable psychological experience. This low-arousal positive emotion has been found to boost creativity [1], organizational citizenship behavior [2], and prosocial behavior [3]. Studies have also found that happiness is negatively associated with mental health (e.g., [4]) and physical health (e.g., [5]). For example, Otsuka and colleagues found a negative relationship between subjective happiness and sleep disturbance [6]. They analyzed responses of 64,152 adolescents and found that subjective happiness has a negative relationship with the prevalence of insomnia, short sleep duration, and poor sleep quality respectively.

While pursuing happiness has been of great interest, it is important to take note that sources of happiness vary. According to the cultural construal of happiness framework [7], individualistic people obtain happiness from achieving individual accomplishments and autonomy, while individuals in collectivist settings may depend on strong relational connections to acquire happiness. Most of the existing measurements of happiness, however, are developed based on the individualistic approach. It is believed that the individualistic perspective may not be able to capture the full picture of happiness. For instance, in their study to examine the definitions of happiness on 2799 adults from 12 countries, Delle Fave and colleagues found that harmony was mentioned more frequently than other individual components, such as meaning and autonomy [8]. Indeed, harmony was the most frequently mentioned component within the psychological category. The findings imply the necessity of investigating happiness using a social orientation approach. Moreover, a new measurement is essential to fulfilling the need of shifting the focus of happiness research. In viewing this need, Hitokoto and Uchida developed the nine-item Interdependent Happiness Scale (IHS) to investigate interpersonal happiness [9].

The IHS assess individuals’ happiness in three dimensions: Relationship-oriented happiness (RH; getting an excellent quality of social contact with other people), quiescent happiness (QH; ability to fulfill expectations of the sociocultural context), and ordinary happiness (OH; achieving a comparable degree of success as others around them). Although Hitokoto and Uchida found IHS score positively correlated with subjective well-being in Japan, Germany, South Korea, and the U.S. samples [9], they did not explicitly examine the psychometric qualities of the IHS.

Datu and colleagues examined psychometric properties of the IHS among Filipino high school students [10]. The results revealed a second-order model, in which the RH, QH, and OH (first-order) factors can be accounted for by an interdependent happiness construct. The latter had a stronger relationship with flourishing and life satisfaction than each dimension (Study 1), at the same time predicting flourishing and life satisfaction even after excluding the sense of relatedness (Study 2). Datu and colleagues administered the IHS thrice (with one-month interval between each data collection) to Filipino secondary school students and tested longitudinal factorial invariance of the IHS [11]. The researchers found that the IHS showed structural (i.e., scalar) invariance. Furthermore, using a three-wave cross-lagged design, Datu and colleagues found that interdependent happiness positively predicts academic engagement measured at Time 2 (i.e., one month after the first survey) and Time 3 (i.e., two months after the first survey) [11].

Although Datu and colleagues showed support to the usefulness of the IHS in the Filipino context [10,11], the mono-cultural investigation does not guarantee the applicability of the IHS in other cultural settings. Indeed, the IHS has been found to have mixed psychometric qualities in different cultural contexts. When applying the IHS to examine the association between age and interdependent happiness in adults across Costa Rica, Japan and the Netherlands, Hitokoto and Takahashi found that the IHS had low reliability for the Netherlands (sample 1) and Costa Rica participants [12]. They further conducted multi-group confirmatory factor analysis (MGCFA) to identify items that are not comparable across the samples. Their results showed that the IHS did not meet measurement unit (i.e., metric) equivalence. The issue, however, was resolved after omitting item 1. The eight-item IHS achieved partial measurement unit equivalence [12].

### The Present Study

Gardiener and colleagues [13] compared the Subjective Happiness Scale (SHS) [14] with IHS across 63 countries and conclude that IHS is more suitable than the SHS to be a universal measurement of happiness. The findings not only emphasize the importance of studying interdependent happiness but also highlight the need of verifying the usefulness of the IHS across different cultures. However, as reviewed above, the psychometric qualities of the IHS are inconsistent in different countries [12]. It is essential to validate the IHS across diverse cultural contexts because even the satisfactory psychometric qualities of a measurement found in a cultural context may not be generalized to other cultural settings (e.g., [1,15,16,17]). Therefore, the present study investigated the psychometric properties and measurement invariance of the IHS in three different countries (Malaysia, Philippines, and India) to clarify the usability of the IHS. Although Gardiener and colleagues collected data for the English version of IHS in the Philippines and India [13], 96.09% of the Malaysian participants in their study answered the Malay, but not English, version of IHS. Hence, reanalysis of Gardiener and colleagues’ open access data is unable to completely fulfill the objective of this research [13].

## 2. Method

### 2.1. Participants

A total of 812 (251 males, 544 females, 17 missing values) undergraduate students from Malaysia (*n* = 263, 101 males and 157 females, five missing values), Philippines (*n* = 239, 74 males and 154 females, 11 missing values), and India (*n* = 310, 76 males and 233 females, one missing value) were recruited using convenience sampling. The mean age of the participants was 21.60 (*SD* = 4.39), ranging from 18 to 64. All participants gave their consent and voluntarily answered the online survey without compensation. The three countries were chosen to meet the requirements of a larger project on creativity. The project was approved by the Scientific and Ethical Review Committee of the Universiti Tunku Abdul Rahman (Ref: U/SERC/107/2018).

### 2.2. Measures

All participants answered the original, English version of the following measurements.

#### 2.2.1. Interdependent Happiness Scale (IHS)

The IHS [9] is a measurement of interpersonal happiness. Individuals respond to the nine items using a 5-point Likert scale ranging from 1 (*strongly disagree*) to 5 (*strongly agree*). Higher mean scores indicate a higher level of interdependent happiness.

#### 2.2.2. Self-Rated Creativity Scale (SRCS)

The SRCS [18] consisted of 12 items (e.g., “*I come up with new and practical ideas to improve performance*”) for respondents to self-report their creativity on a 5-point Likert scale (1: Strongly disagree; 5: Strongly agree). A higher mean score indicates a higher level of (self-perceived) creativity.

### 2.3. Statistical Analysis

Confirmatory factor analysis (CFA) was conducted on each of the three country samples using the R beta module of the JASP ver. 0.16 [19]. The lavaan R package [20] was employed to investigate the factorial structure of the IHS using maximum likelihood robust (MLR) estimation. A total of four models were tested. On top of the conceptual model (i.e., a nine-item second-order model with a general factor of interdependent happiness and three first-order factors) revealed by Hitokoto and Uchida [9], we also tested the one-factor model with nine items, the second-order model with eight items (without item 1) uncovered by Hitokoto and Takahashi [12], and the one-factor model with eight items. A good fit model is identified by the following indices and the suggested cut-off values: Chi-square value to degrees of freedom ratio (χ^2^/df < 3), comparative fit index (CFI > 0.95) and Tucker Lewis Index (TLI > 0.95), and root mean square error of approximation (RMSEA < 0.06), and standardized root mean squared residual (SRMR < 0.08) [21,22]. Modification indices were consulted for suggestions on further improving models whose fit indices are close to the above-mentioned cut-off values. When there is more than one good-fit model, the conceptual model is preferable. In the case that the conceptual model is not one of the good-fit models, we consulted Akaike’s Information Criteria (AIC) and Bayesian Information Criteria (BIC) to identify the best fit model.

We used the JASP and semTools R package [23] to examine the reliability, construct validity (i.e., convergent and discriminant validity), and concurrent validity of the IHS. Reliability was tested using Cronbach alpha and McDonald’s omega coefficients, while construct validity was tested using average variance extracted (AVE). Convergent validity is supported if the AVE value is greater than 0.50, while discriminant validity is evident if the AVE value of the subscale is greater than the squared correlation of the subscale with any other subscale [24]. On the other hand, concurrent validity was tested by correlating the IHS score with the SRCS score because happiness has been found to enhance creativity (e.g., [1]).

When a particular model consistently shows a good fit across the three samples, we will further examine the measurement invariance (MI) of the IHS using MGCFA. The first step is to examine the configural invariance model (i.e., the same factor structure applies to three country groups) and followed by the metric invariance model (i.e., the factor loading pattern is identical across three groups) and scalar invariance model (i.e., intercepts are assumed to be identical across three groups). MI is tested by comparing the models and is supported if ∆CFI < 0.01 and ∆RMSEA < 0.015 [25]. For example, metric invariance is supported when the (absolute value of the) difference between the CFI value of the configural invariance model and the metric invariance model is <0.01. When any of the two indicator values exceed the suggested cut-off (i.e., non-invariance), we explored the possibility of partial invariance by referring to the modification indices for the suggestion of releasing constraint and allowing the factor loading or intercept to be freely estimated.

## 3. Results

Missing values were found in five items (i.e., items 1, 3, 5, 6, and 8) of the IHS and four items (i.e., items 1, 4, 5, and 8) of the SRCS. Missing value analysis showed that the highest percentage of missing was 0.2%. Moreover, the Little’s test was not statistically significant: χ^2^ (72) = 76.539, *p* = 0.335, suggesting that the data were missing at completely random. As a result, we applied expectation maximization to handle the missing values.

### 3.1. Confirmatory Factor Analysis

The three datasets were submitted to CFA respectively to identify the best fit model for each country sample. Table 1 shows the results based on the robust values derived from the correction of MLR estimation for each country. For the Malaysian sample, both one-factor models with nine items (Model MY1) and eight items (Model MY2) showed poor fit. The nine-item second-order model (Model MY3) showed mixed results: CFI and SRMR, but not TLI and RMSEA, values. After consulting the modification indices to add an error covariance between items 7 and 8, the modified model (Model MY3a) showed a good fit. Similarly, the eight-item second-order model (Model MY4) also demonstrated mixed results. The modified model (Model MY4a) with an error covariance between items 7 and 8 showed an excellent fit. Following the planned selection guideline, the Model MY3a is kept for the Malaysian sample.

Similarly, both one-factor models with nine items (Model PH1) and eight items (Model PH2) showed poor fit to the data among the Filipino responses. On the other hand, all fit indices except for TLI showed the second-order model with nine items (Model PH3) had a good fit. The inconsistency was resolved after adding an error covariance between items 3 and 8. However, the (standardized) factor loading of the ordinary happiness (latent) factor was greater than 1 and had a negligible negative variance (standardized value = −0.016). We fixed the variance of the ordinary happiness factor to zero and the modified model (Model PH3a) showed an excellent fit. Like the Malaysian sample, the second-order model with eight items (Model PH4) showed mixed results. However, we did not modify the model because modification indices suggested none of the adjustments can bring a major improvement (i.e., modification index > 10) to the model. As a result, Model PH3a is the best fit model for our Filipino sample.

Finally, the one-factor model with nine items (Model IND1) and eight items (Model IND2) showed poor fit to the Indian data, while the second-order model with nine items (Model IND3) demonstrated mixed results. The latter showed an excellent fit after fixing the variance of the quiescent happiness (latent) factor to zero (due to a negligible negative score) and adding three pairs of error covariances between items 3 and 4, items 3 and 5, and items 5 and 9 (Model IND3a). Likewise, the results for the second-order model with eight items (Model IND4) were inconsistent. Fixing the variance of the quiescent happiness factor to zero and adding two pairs of error covariances between items 3 and 5 and items 3 and 8 successfully improved the modified model (i.e., Model IND4a) to an excellent fit. Model IND3a is selected as the preferable model following the intended selection guideline.

### 3.2. Measurement Invariance across Countries

Although the conceptual nine-item second-order model shows a good fit to each country sample after modification (i.e., adding error covariances), the modified model is different from one sample to another. In other words, a single model that applies to each sample has not been found. Hence, we did not conduct a measurement invariance test in the present study.

### 3.3. Reliability and Validity

Table 2 shows descriptive statistics and reliability coefficients for the three-country samples. The overall IHS and the subscales except for the quiescent happiness subscale showed good internal consistency for the Malaysian sample. The Filipino sample showed unsatisfactory reliability. The overall IHS, but none of the three subscales, had acceptable internal consistency. On the contrary, the overall IHS and all subscales showed good internal consistency for the Indian sample.

Table 3 shows the intercorrelation results and AVE values. The correlation analysis showed that the overall IHS and its three subscales positively correlated with each other across the three-country samples. However, the convergent validity results are mixed. The Malaysian sample only showed a value greater than 0.50 in the ordinary happiness subscale, while the AVE values (of the overall IHS and the three subscales) were all below the cut-off in the Filipino sample. In contrast, the AVE values of the overall IHS and the three subscales were all greater than 0.50 in the Indian sample.

Discriminant validity of the IHS scale was also evident. The AVE value of the overall IHS scale was greater than the squared correlation of the overall IHS score with self-rated creativity score for the three-country samples respectively. Similarly, the AVE values of the three subscales were greater than the squared correlation of itself with the other two subscales respectively in the three-country samples apart from the quiescent happiness subscale in the Indian sample.

The IHS scale also showed good concurrent validity. Both the overall IHS and the three subscale scores positively correlated with the self-rated creativity score, respectively for the three-country samples respectively. The Pearson correlation coefficient ranged from 0.209 (quiescent happiness) to 0.289 (relationship-oriented happiness) for the Malaysian sample, 0.295 (quiescent happiness) to 0.418 (IHS) for the Filipino sample, and 0.574 (quiescent happiness) to 0.665 (relationship-oriented happiness) for the Indian sample.

### 3.4. Supplementary Analysis

Both Model MY3a and Model MY4a showed excellent fit to the Malaysian data. Although the former is preferable, it is intriguing to know if both two models are applicable (or one is superior to the other). We compared the AIC, BIC, and reliability of the two models. Although Model MY4a (AIC = 5088.537, BIC = 5159.980) had a lower value in both indices than Model MY3a (AIC = 5650.787, BIC = 5729.374), the reliability coefficients of the relationship-oriented happiness subscale of Model MY4a (with 2 items; α = 0.688, ω = 0.693) were also lower than Model MY3a (with 3 items; α = 0.728; ω = 0.730).

Similarly, we also compared the Model IND3a and Model IND4a of the Indian sample. The latter (AIC = 4652.865, BIC = 4727.596) showed a lower value than Model IND3a (AIC = 5212.195, BIC = 5298.136) in both indices. Notably, the reliability coefficients (for the overall IHS and subscale scores) of the two models are compatible and all above 0.70. Appendix A shows the standardized factor loadings and reliability coefficients for Model MY4a and Model IND4a.

Appendix A shows the intercorrelation results and AVE values for Model MY4a and Model IND4a. The construct validity (i.e., convergent validity and discriminant validity) and concurrent validity results are consistent with the results for Model MY3a and Model IND3a.

## 4. Discussion

The nine-item Interdependent Happiness Scale (IHS) [9] with three subscales is one of the few happiness measurements based on the collectivistic approach. The present study is the first to shed light on the psychometric qualities of the IHS and measurement invariance across three Asia countries (i.e., Malaysia, Philippines, and India). The results reveal that the psychometric properties of the IHS vary from one sample to another.

The results of the CFA supported that the IHS is best interpreted as a higher-order construct that consists of three first-order dimensions: Relationship-oriented happiness, quiescent-oriented happiness, and ordinary-oriented happiness in the three samples. The findings are consistent with Datu and colleagues′ results obtained from Filipino high school students [10]. Although the nine-item higher-order factorial structure applies to the three samples, it is important to highlight different error covariances that had been imposed to improve the model fit for each sample. For example, error covariance was added between items 7 (“*I believe that my life is just as happy as that of others around me*”) and 8 (“*I believe I have achieved the same standard of living as those around me*”) in the Malaysia sample. The modification suggests that the two items are perceived conceptually similar to each other by the Malaysian participants. This is reasonable as one’s happiness can be a part of the standard of living. Likewise, items 3 and 8 are conceptually parallel for the Filipino sample, while the Indian sample comprehended three pairs of items (items 3 and 4, items 3 and 5, and items 5 and 9) as conceptually compatible. In other words, the results imply that there could be cultural differences in some items of the IHS. Future studies are warranted to examine the meaning of the items qualitatively and then revise the items to enhance the compatibility and usability in diverse cultures.

The present study failed to identify a common factorial structure that holds for every sample. As a result, the measurement invariance of the IHS across three countries was not tested. The results are contradictory with the literature where IHS was found to have partial metric invariance across Costa Rica, Japan, and the Netherlands after allowing freely estimation on item 1 [12] and structural invariance (i.e., factor variances and covariances are constrained to be equal) on secondary school students in the Philippines [11]. Moreover, some items do not hold across samples. For example, consistent with the findings of Hitokoto and Takahashi from Costa Rica, Japan, and the Netherlands [12], the eight-item second-order model without item 1 showed a good fit to the Malaysian and Indian samples (after adding error covariance). The same pattern, however, was not observed in the Filipino sample. Future researchers are recommended to investigate the factorial structure of the IHS before applying it in their target samples.

The overall IHS displayed good reliability in the three-country samples respectively. The results are consistent with the findings of Gardiner and colleagues measured with McDonald’s omega total [13]. However, the reliability of the three subscales was inconsistent among the three samples. All the subscales had good internal consistency in the Indian sample but showed unsatisfactory results in the Filipino sample. The unsatisfactory internal consistency implies that the items of each subscale seem to be unrelated to each other to the Filipino participants. For the Malaysian sample, the reliability of all subscales was satisfactory except for the quiescent happiness subscale. Further inspection (on the Malaysian sample) showed that item 4 (“Although it is quite average, I live a stable life”) and item 6 (“I can do what I want without causing problems for other people”) had the lowest inter-item correlation (0.353). The results imply that the two items of the quiescent happiness subscale carry somewhat different meanings to the Malaysian participants. Future researchers may consider modifying the items to enhance their clarity and strengthen the relatedness of the items if the inadequate reliability is replicated.

Construct validity (i.e., convergent and discriminant validity) and concurrent validity of the IHS scale was examined in the present study. Convergent validity was demonstrated in the Indian sample, but not the Filipino and Malaysian samples (except for the ordinary happiness subscale). On the other hand, discriminant validity was evident in the three-country samples respectively for the overall IHS scale and the subscales except for the quiescent happiness subscale in the Indian sample. The results of construct validity indicate that the items do not seem closely related with each other to the Filipino and Malaysian participants, while Indian participants are unable to distinguish the quiescent happiness subscale from the other two subscales. On the other hand, concurrent validity of the IHS scale was established. The overall IHS scale and the three subscales were positively correlated with the self-reported creativity in the three sub-samples respectively. The results are consistent with past findings that happiness is conducive to creativity [1].

Our findings contribute to the literature by providing empirical evidence to the psychometric qualities of the IHS across three countries. The results not only provide empirical evidence to the strengths and weaknesses of the IHS in the Asian context but also serve as a useful reference for researchers who wish to measure interdependent happiness among young adults across different cultures especially in the context of Malaysia, the Philippines, and India. For instance, our supplementary analysis results imply that removing item 1 (“I believe that I and those around me are happy”) has little negative impact on the Indian sample than the Malaysian and Filipino samples. Put differently, item 1 is perceived as essential to Malaysian and Filipino participants (especially the latter) but not Indian participants. It is a promising direction for future studies to examine the item qualities quantitatively (using item response theory analysis) and qualitatively.

Although the results are promising, it is noteworthy that the present study only focused on three countries in Asia. Moreover, the preferable nine-item second-order model established a good fit after adding error covariances. Note that such findings derived from model modifications are exploratory and require further confirmation. Therefore, it is premature to conclude that the results can be generalized to other countries. In the same vein, the present study has a narrowed focus on university students. The results derived from young adults may not apply to other age groups. Finally, the scope of the investigation is limited. Test-retest reliability and predictive validity were not examined in the present study. More studies are warranted to address these limitations by replicating the study in diverse cultural contexts, extending the investigation to other populations with larger sample size, and applying a longitudinal design. Moreover, future researchers may correlate the IHS score with other established measurements of happiness or well-being, such as the five-item World Health Organization Well-Being Index (WHO-5) that have been validated in different countries including Malaysia [26] to clarify the mixed findings of construct validity. It is also suggested to employ a qualitative approach to ensure the items (especially item 1) are adequate in the target cultural environment. Lastly, future researchers are suggested to investigate the antecedent factors of interdependent happiness and explore the possible causes of the differences in interdependent happiness. For instance, autonomous motivation and academic engagement have been found to predict the interdependent happiness of secondary school students [11]. Meanwhile, Tan and colleagues [27,28] found that self-esteem, social support, and hope are positively associated with the happiness of young adults in Malaysia. It is a potential direction to examine if the low level of interdependent happiness observed in the Malaysian sample is attributed to the above-mentioned factors.

## 5. Conclusions

Across three different cultural samples, it has been found that the nine-item Interdependent Happiness Scale (IHS) is accounted for by a second-order model with a general interdependent happiness factor and three first-order specific factors. The preliminary evidence suggests that the IHS has the potential to capture the concept of socially oriented happiness. However, the psychometric qualities (e.g., reliability & validity) of the IHS vary from one sample to another. Further investigations on the item qualities are warranted.

## Figures and Tables

**Table 1 ijerph-19-00187-t001:** The Fit Indices for the Models of the Interdependent Happiness Scale.

Model		χ^2^	df	*p*	χ^2^/df	CFI	TLI	RMSEA [90% CI]	SRMR
**Malaysia**									
MY1	One-factor with 9 items	90.947	27	<0.001	3.368	0.902	0.869	0.103 [0.080, 0.126]	0.059
MY2	One-factor with 8 items	75.727	20	<0.001	3.786	0.896	0.854	0.112 [0.086, 0.140]	0.062
MY3	2nd-order model with 9 items	51.591	24	0.001	2.150	0.958	0.937	0.071 [0.044, 0.098]	0.045
MY3a	2nd-order model with 9 items with error covariance ^a^	40.415	23	0.014	1.757	0.973	0.958	0.058 [0.026, 0.087]	0.040
MY4	2nd-order model with 8 items (removed item 1)	34.195	17	0.008	2.011	0.968	0.948	0.067 [0.033, 0.100]	0.042
MY4a	2nd-order model with 8 items with error covariance ^a^	22.072	16	0.141	1.380	0.989	0.980	0.041 [0.000, 0.080]	0.033
**Philippines**									
PH1	One-factor with 9 items	55.955	27	0.001	2.072	0.916	0.888	0.075 [0.047, 0.103]	0.058
PH2	One-factor with 8 items	43.645	20	0.002	2.182	0.916	0.882	0.078 [0.046, 0.110]	0.056
PH3	2nd-order model with 9 items	36.275	24	0.052	1.511	0.965	0.948	0.051 [NA, 0.083]	0.046
PH3a	2nd-order model with 9 items with error covariance ^b^	27.773	24	0.270	1.157	0.989	0.984	0.028 [0.000, 0.067]	0.042
PH4	2nd-order model with 8 items (removed item 1)	31.296	17	0.018	1.841	0.951	0.920	0.064 [0.026, 0.099]	0.046
**India**									
IND1	One-factor with 9 items	80.176	27	<0.001	2.969	0.920	0.893	0.142 [0.107, 0.179]	0.051
IND2	One-factor with 8 items	68.951	20	<0.001	3.448	0.914	0.880	0.159 [0.119, 0.201]	0.054
IND3	2nd-order model with 9 items ^c^	51.007	25	0.002	2.040	0.962	0.945	0.102 [0.061, 0.142]	0.041
IND3a	2nd-order model with 9 items with error covariance ^d^	26.002	22	0.252	1.182	0.994	0.990	.043 [0.000, 0.098]	0.026
IND4	2nd-order model with 8 items (removed item 1) ^c^	42.976	18	0.001	1.953	0.956	0.931	0.120 [0.074, 0.167]	0.044
IND4a	2nd-order model with 8 items with error covariance ^e^	20.484	16	0.199	1.280	0.992	0.986	0.054 [0.000, 0.116]	0.035

The reported indices were based on robust values corrected in accordance with the MLR estimator. TLI = Tucker-Lewis Index, CFI = comparative fit index, RMSEA = root-mean-square error of approximation, CI = confidence interval, SRMR = standardized root mean square residual. ^a^ Added error covariance between items 7 and 8; ^b^ Added error covariance between items 3 and 8 and fixed the variance of the Ordinary Happiness to zero; ^c^ Fixed the variance of Quiescent Happiness to zero; ^d^ Fixed the variance of the Quiescent Happiness to zero and added error covariance between items 3 and 4, items 3 and 5, and items 5 and 9; ^e^ Fixed the variance of Quiescent Happiness to zero and added error covariance between items 3 and 5 and items 3 and 8.

**Table 2 ijerph-19-00187-t002:** Factor loading, descriptive statistics, and reliability of the Interdependent Happiness Scale with nine items.

	Malaysia (*n* = 263)					Philippine (*n* = 239)					India (*n* = 310)				
Item/Factor	FL	*M*	*SD*	*α*	ω	FL	*M*	*SD*	*α*	ω	FL	*M*	*SD*	*α*	ω
Relationship-oriented happiness	0.857	3.68	0.67	0.728	0.730	0.806	3.82	0.58	0.676	0.676	0.923	3.85	0.78	0.876	0.878
IHS1	0.700	3.59	0.87			0.667	3.85	0.83			0.819	3.77	0.94		
IHS2	0.704	3.60	0.85			0.662	3.73	0.72			0.879	3.85	0.86		
IHS3	0.658	3.83	0.78			0.594	3.86	0.68			0.829	3.94	0.80		
Quiescent happiness	0.884	3.47	0.76	0.652	0.641	0.837	3.71	0.67	0.550	0.557	1.00	3.79	0.78	0.792	0.829
IHS4	0.650	3.90	0.85			0.486	3.97	0.83			0.773	3.94	0.81		
IHS5	0.586	3.00	1.11			0.538	3.43	1.02			0.747	3.57	1.10		
IHS6	0.627	3.50	1.02			0.601	3.72	0.91			0.806	3.86	0.85		
Ordinary happiness	0.954	3.35	0.83	0.795	0.708	1.00	3.67	0.65	0.637	0.674	0.932	3.72	0.84	0.901	0.893
IHS7	0.729	3.28	1.00			0.708	3.67	0.86			0.859	3.71	0.91		
IHS8	0.633	3.22	1.01			0.696	3.59	0.90			0.877	3.71	0.89		
IHS9	0.769	3.54	0.94			0.453	3.75	0.77			0.858	3.74	0.96		
nine-item IHS	-	3.50	0.63	0.849	0.858	-	3.73	0.51	0.788	0.816	-	3.79	0.74	0.938	0.943

FL = standardized factor loading; *M* = mean; *SD* = standard deviation; *α* = Cronbach alpha coefficient; ω = McDonald omega coefficient; IHS: Overall interdependent happiness scale score.

**Table 3 ijerph-19-00187-t003:** Correlation squared correlation, and average variance extracted from the Interdependent Happiness Scale.

	1	2	3	4	5	AVE
**Malaysia**						
1.IHS	1	0.656	0.691	0.757	0.077	0.450
2.Relationship	0.810 ***	1	0.258	0.334	0.084	0.475
3.Quiescent	0.831 ***	0.508 ***	1	0.325	0.044	0.380
4.Ordinary	0.870 ***	0.578 ***	0.570 ***	1	0.043	0.504
5.Creativity	0.277 ***	0.289 ***	0.209 **	0.208 **	1	-
**Philippines**						
1.IHS	1	0.593	0.663	0.716	0.175	0.368
2.Relationship	0.770 ***	1	0.167	0.263	0.100	0.418
3.Quiescent	0.814 ***	0.409 ***	1	0.286	0.087	0.299
4.Ordinary	0.846 ***	0.513 ***	0.535 ***	1	0.164	0.413
5.Creativity	0.418 ***	0.316 ***	0.295 ***	0.405 ***	1	-
**India**						
1.IHS	1	0.828	0.885	0.867	0.413	0.681
2.Relationship	0.910 ***	1	0.626	0.561	0.442	0.708
3.Quiescent	0.941 ***	0.791 ***	1	0.692	0.329	0.594
4.Ordinary	0.931 ***	0.749 ***	0.832 ***	1	0.308	0.747
5.Creativity	0.643 ***	0.665 ***	0.574 ***	0.555 ***	1	-

AVE: Average variance extracted; IHS: Overall interdependent happiness scale score; Relationship: Relationship-oriented happiness. Above diagonal line is the squared correlation coefficient. ** *p* < 0.01, *** *p* < 0.001.

## Data Availability

The datasets generated during and/or analyzed during the current study are available from the corresponding author on reasonable request.

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
