# Peer review of "Psychometric Qualities Evaluation of the Interdependent Happiness Scale across Malaysia, Philippines, and India"

_ijerph, 2021, doi:10.3390/ijerph19010187_

Round 1

Reviewer 1 Report

The authors attempted to validate the 9-item Interdependent Happiness Scale in Malaysia (n = 263), Philippines (n = 239), and India (n = 310) with 812 samples in total. However, there are several major issues need to be addressed by the authors:

1) Please report the translation procedure in the methods section.

2) The SRCS was validated in the contexts of Malaysia, Philippines and India?

3) Which estimator was used in the CFA? Please note that there are recent literature discussed the appropriateness of the CFA estimator (Li, 2016).

4) Please also report the software and version used to compute the data

5) In addition to the Cronbach’s alpha, please also report the McDonald’s Omega (Beland, Cousineau, & Loye, 2017).

6) The CFA result in the Indian sample “χ2 (24) = 148.808, p < .001, χ2/df = 6.200, CFI = .943, TLI = .914, RMSEA = .130 [.110, .150], 178 SRMR = .040” (p. 4, lines 178-179) did not fulfill the criteria of adequate model fit in RMSEA and chi-square/df. Please further discuss and elaborate it.

7) Some of the sub-scales reported on Table 1 also failed to fulfil the criteria for Cronbach alpha value > 0.70. The proposed three-latent factor structure maybe a bit problematic. The authors may need to further re-analyse the data and compare the current proposed version with those variants proposed in the existing Happiness Scale literature, such as the 8-item version, etc.

8) For Table 3, the authors also need to show the results separately based on the samples in Malaysia, Philippines, and India.

9) In the conclusion, I am afraid that the reported results did not support the following claims “The 9-item Interdependent Happiness Scale (IHS) shows sound psycho-metric qualities including measurement invariance across three different countries” (p. 12).

References

Beland, S., Cousineau, D., & Loye, N. (2017). Using the Mcdonald's Omega Coefficient instead of Cronbach's Alpha. Mcgill Journal of Education, 52(3), 791-804. Retrieved from <Go to ISI>://WOS:000437472500013

Li, C. H. (2016). Confirmatory factor analysis with ordinal data: Comparing robust maximum likelihood and diagonally weighted least squares. Behavior Research Methods, 48(3), 936-949. doi:10.3758/s13428-015-0619-

Reviewer 2 Report

- In the Introduction it is not clear the objective of this manuscript. Authors say that: 

the present study investigated the psychometric properties and measurement invariance of the IHS in three different countries (Malaysia, Philippines, and India) to clarify the usability of the IHS. 

Reading Introduction seems that they want to verify this question using english version in all samples, but it is not clear.

  • When measures are presented is not explained the language used for the instruments

-In Confirmatory Analysis that CFA results supported the model for all samples, however for India is presented a chi/df:6.2 and RMSEA: .130 (.110, .150) These values exceed the cut-off values (chi/df <3 and RMSEA <0.08). So the model does not adjust in an adequate manner for India.

- Reliability and Validity: Authors  wrote that results of Reliability showed good reliability, however they also wrote that some parts had low values either in a sample, or in a construct. So reliability is not as good as they affirm. Reliability needs values greater than .70 to be considerate good.

- Measurement  Invariance across Countries. The methodology followed for test invariances across countries is not adequate, due that authors considered all three samples in the same analyses, so it is not possible establish if equivalences, or differences, are between Malaysia and Philippine; Malaysia and India or Philippine and India. So it is necessary to run analyses 3 different analyses to be sure were are the equivalences or differences. 

- After that ANCOVAS are made, however, when invariance is achieved, latent means analyses are better for checking the differences, I recommend follow:

Dimitrov, D. (2010). Testing for factorial invariance in the context of construct validation. Measurement and Evaluation in Counseling and Development, 43, 121–149. doi: 10.1177/0748175610373459

Round 2

Reviewer 1 Report

Thanks for submitting the revised manuscript to address my comments. I am fully satisfied with the responses and changes made by the authors. There is only one outstanding issue in point number 6:

6) Please provide the justifications for correlating the error terms (Hermida, 2015).

References

Hermida, R. (2015). The problem of allowing correlated errors in structural equation modeling: concerns and considerations. Computational Methods in Social Sciences (CMSS), 3(1), 05-17. Retrieved from https://EconPapers.repec.org/RePEc:ntu:ntcmss:vol3-iss1-15-005

Author Response

Thank you for the further comment.

Methodologically speaking, correlating the error terms is a useful method to explore the possible ways to improve the model fit (Byrne, 2016). 

We provided the theoretical justifications on p. 14-15 (see the part highlighted in green). For instance, error covariance was added between items 7 (“I believe that my life is just as happy as that of others around me”) and 8 ("I believe I have achieved the same standard of living as those around me”) in the Malaysia sample because the two items are perceived conceptually similar to each other by the Malaysian participants. This is reasonable as one’s happiness can be a part of the standard of living. 

Byrne, B. M. (2016). Structural Equation Modeling with AMOS Basic Concepts, Applications, and Programming (3rd Edition). Routledge.